



# Distribution of the Earth's radiation belts protons over the drift frequency of particles

**Alexander S. Kovtyukh**

Skobeltsyn Institute of Nuclear Physics, Moscow State University, Moscow, 119234, Russia

Correspondence: Alexander S. Kovtyukh (kovtyukhas@mail.ru)

**Abstract.** On the base of generalized data on the proton fluxes of the Earth's radiation belts (ERB) with energy from $E \sim 0.2$ MeV to 100 MeV at drift shells $L$ from $\sim 1$ to 8, constructed stationary distributions of the ERB protons over the drift frequency $f_d$ of protons around the Earth. For this, direct measurements of proton fluxes of the ERB in the period 1961–2017 near the plane of the geomagnetic equator were used. The main physical processes in the ERB manifested more clearly in these distributions, and for protons with $f_d > 0.5$ mHz at $L > 3$ distributions of the ERB protons in the space $\{f_d, L\}$ have a more orderly form than in the space $\{E, L\}$. It has been found also that the quantity of the ERB protons with $f_d \sim 1$–10 mHz at $L \sim 2$ does not decrease, as for protons with $E > 10$–20 MeV (with $f_d > 10$ mHz), but increases with an increase in solar activity. This means that the balance of radial transport and losses of the ERB low-energy protons at $L \sim 2$ is disrupt in advantage of transport: for these protons, the effect of an increase in the radial diffusion rates with increasing in solar activity overpowers the effect of an increase in the density of the dissipative medium.

**Keywords**. Magnetospheric physics (energetic particles, trapped). Radiation belts.





## 1 Introduction

The Earth's radiation belts (ERB) consist mainly of charged particles with energy from $E \sim 100$ keV to several hundreds of megaelectronvolt (MeV). In the field of the geomagnetic trap, each particles of the ERB with energy $E$ and equatorial pitch-angle $\alpha_0$ ($\alpha$ is the angle between the local vector of the magnetic field and the vector of a particle velocity) makes three periodic movements: Larmor rotation, oscillations along the magnetic field line, and drift around the Earth (Alfvén and Fälthammar, 1963; Northrop, 1963).

Three adiabatic invariants ($\mu$, $K$, $\Phi$) correspond to these periodic motions of trapped particles, as well as three periods of time or three frequencies: a cyclotron frequency $f_c$, a frequency of particle oscillations along the magnetic field line $f_b$, and a drift frequency of particles around the Earth $f_d$. For the near-equatorial ERB protons, these frequencies belong to the following ranges: $f_c$ $\sim 1$–500 Hz, $f_b \sim 0.02$–2 Hz and $f_d \sim 0.1$–20 mHz. The frequency $f_c$ increases by tens to hundreds of times with the distance of the particle from the plane of the geomagnetic equator (in proportion to the local induction of the magnetic field), and the frequency $f_b$ decreases by almost 2 times with increasing the amplitude of particles oscillations.

The frequency $f_c$ is different for different $L$-shells (near the equatorial plane) and as $L$ increases it refers to an insignificant number of particles at higher and higher geomagnetic latitudes. Each given value of the frequency $f_b$ with increasing $L$ correspond to particles of more and more higher energies ($E \propto L^2$) and it value encompass fewer and fewer particles.

Compared to the frequencies $f_c$ and $f_b$, the drift frequency $f_d$ of the ERB particles of one species belongs to a much narrower range; the frequency $f_d$ does not depend on the mass of particles and very weakly depends on the amplitude of their oscillations (vary within $\sim 20\%$). Herein, on each $L$-shell of the ERB there are a significant number of particles corresponding to a certain value of $f_d$ from a narrow frequency range.

Therefore, it can be expected that the distributions of the ERB particles in the space $\{f_d, L\}$ will have a more orderly shape than in the space $\{E, L\}$, and the main physical processes in the ERB will manifest themselves more clearly in these distributions. It can also be expected that on these more ordered background will reveal more fine features of the ERB that do not appear in the space $\{E, L\}$.

Meanwhile, despite the importance of the drift frequency $f_d$ for the mechanisms of the ERB formation, reliable and sufficiently complete distributions of the ERB particles over the frequency $f_d$ have not been presented and these distributions have not been analyzed. This is the first time this is done here.

For greater reliance, this analysis is limited here to the protons of the ERB and it is refer to the magnetically quiet periods of observations, when the fluxes of the ERB protons and their spatial-energy distributions were stationary.

In the following sections, the distributions of the ERB protons over their drift frequency $f_d$ were constructed by the experimental data (Sect. 2), and these distributions were analyzed (Sect. 3). Finally, the main conclusions of this work are given in Sect. 4.

## 2 Constructing the distributions of the ERB protons over their drift frequency

### 2.1 Spatial-energy distributions of the ERB protons near the equatorial plane

To construct the distributions of the ERB particles over the drift frequency, it is necessary to have reliable distributions of the differential fluxes of the ERB protons in the space $\{E, L\}$, where $E$ is the kinetic energy of protons and $L$ is the drift shell parameter.



67       According to the data of generalized and averaged satellite measurements of the differential
fluxes of protons with an equatorial pitch-angle $\alpha_0 \approx 90^\circ$, such distributions of proton fluxes for
quiet conditions is constructed in (Kovtyukh, 2020). Such distributions, separately for the periods
near minima and near maxima of the 11-year cycles of solar activity, is constructed from the
satellite data also for other main ionic components of the ERB near the plane of the geomagnetic
equator, but the most reliable and detailed picture was obtained in (Kovtyukh, 2020) for a protons.
In Fig. 1 one of these distributions is reproduced, for periods near maxima of the solar activity
(from 1968 to 2017).
75       Data of satellites are associated in Fig. 1 with different symbols. The numbers on the curves
(iso-lines) refer to the values of the decimal logarithms of the differential fluxes $J$ (cm$^2$ s sr MeV)$^{-1}$
of protons (with equatorial pitch-angle $\alpha_0 \approx 90^\circ$). The red lines in Fig. 1 corresponds to the
dependences $f_d$(mHz) $= 0.379 \cdot L \cdot E$(MeV) for the drift frequency of the near-equatorial protons in the
dipole approximation of the geomagnetic field.

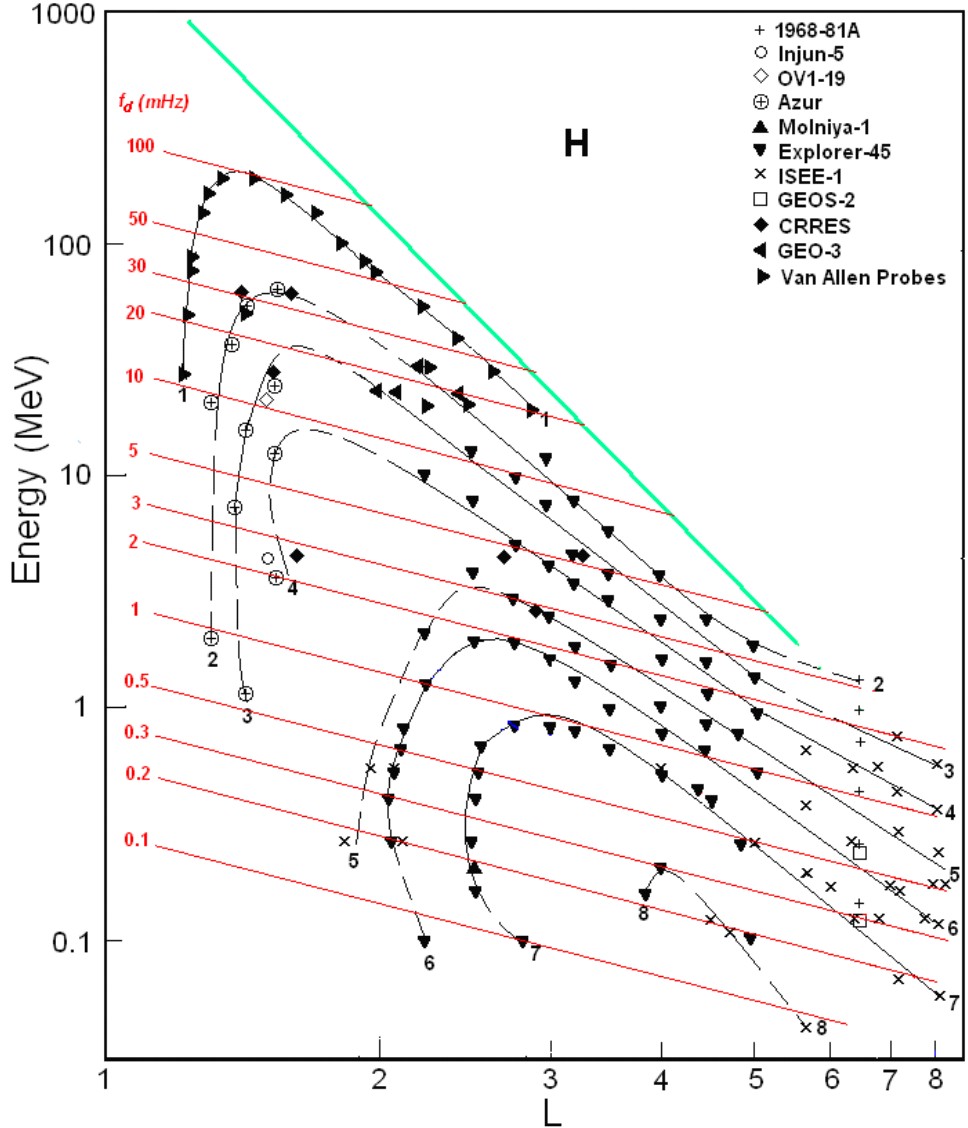






**Figure 1.** Distribution of the differential fluxes $J(E, L)$ in the space $\{E, L\}$ for protons with $\alpha_0 \approx 90^o$ near maxima of
the solar activity (Kovtyukh, 2020). Data of satellites are associated with different symbols. The numbers on the
curves refer to the values of the decimal logarithms of $J$. Fluxes is given in units of $(\text{cm}^2 \text{ s sr MeV})^{-1}$. The red lines
corresponds to the drift frequency $f_d$(mHz). The green line corresponds to the maximum energy of the trapped protons.
On the drift shells can be trapped only protons with energies less than some maximum values,
determined by the Alfvén's criterion: $\rho_c(L,E) << \rho_B(L)$, where $\rho_c$ is the gyroradius of protons, and
$\rho_B$ is the radius of curvature of the magnetic field (near the equatorial plane). According to this
criterion and to the theory of stochastic motion of particles, the geomagnetic trap in the dipolar
region can capture and durably hold only protons with $E$ (MeV) $< 2000 \cdot L^{-4}$ (Ilyin et al., 1984). The
green line in Fig. 1 represents this boundary.
The distribution of the ERB proton fluxes shown in Fig. 1, refer to the years of the solar
maximum, but the solar-cyclic variations in the ERB proton fluxes are small and localized at $L <$
2.5 (mainly at $L < 1.4$).

## 2.2 Spatial-energy distributions of the ERB protons outside the equatorial plane

The stationary fluxes $J$ of the ERB particles with given energy and local pitch-angle $\alpha$ decrease
usually when the point of observation is shifted from the equatorial plane to higher latitudes along
a certain magnetic field line. In the inner regions of the ERB, on $L < 5$, an angular distributions of
protons have usually a maximum at the local pitch-angle $\alpha = 90^o$. In wide interval near this
maximum these distributions are well described by the function
$J(\alpha, B/B_0) \propto (B/B_0)^{-A/2} \sin^A \alpha$ (Parker, 1957), where $A$ is the index of an anisotropy of a
fluxes, $B$ is the induction of a magnetic field at the point of measurements of these fluxes and $B_0$ is
induction of a magnetic field at the equatorial plane on the same magnetic line.

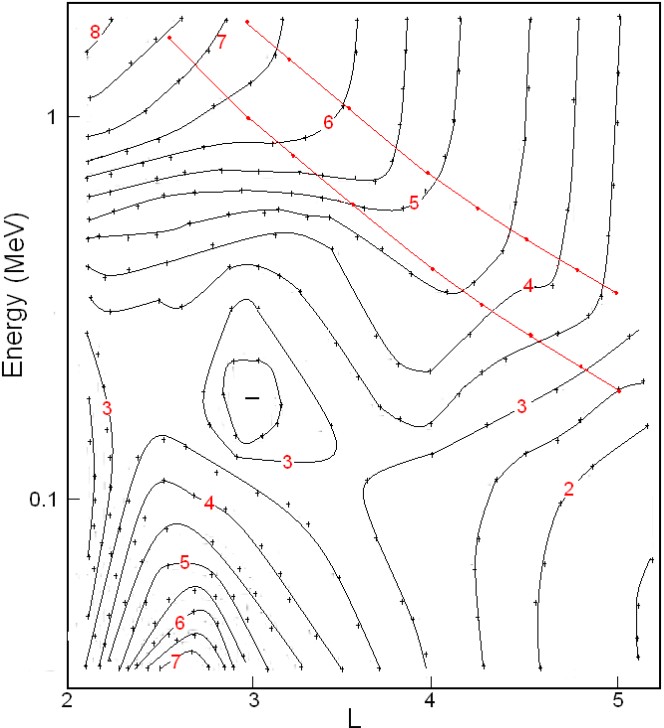






**Figure 2.** Empirical model of the anisotropy index $A(E, L)$ of the ERB proton fluxes averaged on the data of the satellites obtained near the plane of the geomagnetic equator. Values of $A$ are given on iso-lines of the anisotropy: $A = 1.5$–$8.5$ with the step $\Delta A = 0.5$.

The generalized empirical model of an anisotropy $A(E, L)$ for the proton fluxes with $E \sim 0.1$–$2$ MeV on $L \sim 2$–$5$ near the equatorial plane for the stationary ERB ($Kp < 2$) is presented in Fig. 2. The anisotropy index $A$ of the proton fluxes is shown in Fig. 2, in the space $\{E, L\}$, in the form of iso-lines with the same values $A$ from 1.5 to 8.0 and with a step $\Delta A = 0.5$. The integer values of this index are plotted on the corresponding iso-lines in red numbers.

When constructing this model, we consider and analyze the data of the following satellites: Explorer-12 (Hoffman and Bracken, 1965), Explorer-14 (Davis, 1965), Explorer-26 (Søraas and Davis, 1968), Electron (Ilyin et al., 1984; Vlasova et al., 1984), OV1-14 and OV1-19 (Fennell et al., 1974), Explorer-45 (Williams and Lyons, 1974; Fritz and Spjeldvik, 1981; Garcia and Spjeldvik, 1985), Molniya-1 (Vlasova et al., 1984), ISEE-1 (Garcia and Spjeldvik, 1985; Williams and Frank, 1984), SCATHA (Blake and Fennell, 1981), Van Allen Probes (Shi et al., 2016), and other satellites. These data were obtained in 1961-2015.

Figure 2 shows that for rather high energy (> 1 MeV) the anisotropy of a proton fluxes monotonically increases with decreasing $L$ (from $A \sim 3.5$ to $A \sim 8.0$). For $E > 0.3$ MeV on $L < 3$ anisotropy is monotonically increases with increasing energy, but for $E > 0.5$ MeV on $L > 3$ it is almost independent on energy.

Some small irregularities of the iso-lines in Fig. 2 are connect to the fact that experimental data were used for constructing this figure; these data were obtained in different years, with different instruments on different orbits of satellites, and during different intensity of the solar activity. At the same time, Fig. 2 demonstrates the important regularities of the pitch-angle distributions of the stationary ERB protons.

In the region $\{E > 0.5$ MeV, $L > 3\}$ the iso-lines of the anisotropy index are almost parallel to each other and to the energy scale. This adiabatic regularity refers for protons belonging to the power-law tail of their energy spectra, the exponent of which practically does not change when $L$ changes (at $L > 3$). In Fig. 2, the red lines correspond to the lower boundary of the power-law tail of the ERB protons energy spectra: $E_b = (36\pm11)\,L^{-3}$ MeV (see Kovtyukh, 2001, 2020).

The pattern of $A(E, L)$ in the region on $L > 3$ at $E \sim 0.2$–$0.5$ MeV and the local minimum at $L \sim 3$ ($E \sim 0.2$ MeV) are connected with local maximum in the stationary proton energy spectra of the ERB which corresponds to $E = (17\pm3)\,L^{-3}$ MeV (see Kovtyukh, 2001, 2020).

These regularities in the pattern of $A(E, L)$ are explained within the framework of the theory of radial transport (diffusion) of the ERB protons with conservation of the adiabatic invariants $\mu$ and $K$ of their periodic motions (these questions were most fully considered in Kovtyukh, 1993).

Local maximum at $L \sim 2.5$ ($E < 0.1$ MeV) and the region of low anisotropy at $L \sim 2$ ($E \sim 0.1$ MeV) in Fig. 2, are connected with the ionization losses of protons.

High anisotropy for the fluxes of protons at $E = 5$–$50$ MeV and a strong dependence $A(L)$ at the inner boundary of the inner belt ($L = 1.15$–$1.40$, $B/B_0 = 1.0$–$1.7$) were obtained on the satellite DIAL (Fischer et al., 1977). According to these data, an anisotropy index increase from $A \sim 12$ at $L = 1.25$ to $A \sim 60$ at $L = 1.15$, and do not depends on $L$ at $L = 1.25$–$1.40$. These results are supported by the data of the satellite Resurs-01-N4 for the protons with $E = 12$–$15$ MeV which obtained at $h \sim 800$ km (Leonov et al., 2005). They will be taken into account in our calculations.

The experimental results on the pitch-angle distributions of the ERB proton fluxes and their anisotropy indexes were discussed in detail in (Kovtyukh, 2018).

## 2.3 Drift frequency distributions of the ERB protons

Based on the results shown in Fig. 1 and 2, one can calculate the distributions of the ERB protons over the drift frequency $f_d$. In these calculations, the dipole model of the geomagnetic field was



used, according to which (see, e. g., Roederer, 1970) the point of the magnetic field line at
geomagnetic latitude $\lambda$ is located from the center of the dipole at a distance

$$R(L,\lambda) = R_E L \cos^2 \lambda,$$


where $R_E$ is the Earth's radius, and the field induction at a given $L$ changes with changing $\lambda$ as

$$B(L,\lambda) = \frac{\sqrt{4 - 3\cos^2 \lambda}}{\cos^6 \lambda} B_0(L),$$


where $B_0(L) = 0.311$ Gs $\times L^{-3}$.
It was also taken into account that the drift frequency $f_d$ of the nonrelativistic particles depends
essentially only on their kinetic energy $E$ and on $L$. This value depends very slightly on the particle
pitch-angle: with an increase in the geomagnetic latitude of the mirror point of the particle trajectory
from 0 to $10^o$ it increases by only 1.5%, and in the range from 0 to $20$–$30^o$ it increases by 5.8–12.5%.
The number of protons with energies from $E$ to $E+dE$ per unit volume $n$ is equal to the differential
flux of these particles $J$ (falling per unit time per unit area of the detector per unit solid angle), divided
by the velocity $v$ of these particles: $n = J/v$. For nonrelativistic protons with mass $m$, this velocity is
$(2E/m)^{1/2}$.
Then in the near-equatorial region, between $L$ and $L+dL$ and within geomagnetic latitudes from
0 to $\pm\lambda_0$, the total number of nonrelativistic protons with mirror points within this region and with
energy from $E$ to $E+dE$, drifting on a given $L$ with frequency $f_d(L,E)$ around the Earth, is

$$\Delta N(L, f_d) = 2\int_0^{\lambda_0} 2\pi R_E^2 L\, dL\, \frac{B_0(L)}{B(L,\lambda)}\, R_E L\, \cos\lambda\, \sqrt{4 - 3\cos^2\lambda}\, d\lambda \times$$


$$4\pi \int_{\alpha_{01}}^{\alpha_{02}} \frac{J(L, E(L,f_d))dE}{\sqrt{2E(L,f_d)/m}}\, \sin^A \alpha_0\, \cos\alpha_0\, d\alpha_0$$

where $m$ is the rest mass of a proton, $J(L,E(L,f_d))$ is the differential fluxes and $E(L,f_d)$ is the protons
energy. The first integral takes into account that the magnetic flux in the layer between shells $L$ and
$L+dL$ it conserved when latitude $\lambda$ changes, i. e. $2\pi R_E L \cos\lambda\, R_E\, dL = 2\pi R_E L\, \frac{B_0(L)}{B(L,\lambda)}\, R_E\, dL$.
As result of integrating the last expression over $\alpha_0$ and replacing $\cos\lambda \equiv t$, we obtain:

$$\Delta N(L, f_d) = 4\pi R_E^3 L^2 dL\, \frac{J(L, E(L,f_d))dE}{\sqrt{2E(L,f_d)/m}} \times \frac{4\pi}{A+1} \times$$


$$\int_{\cos\lambda_0}^1 t^7 \left[ \left( \frac{t^6}{\sqrt{4 - 3t^2}} \right)^{\frac{A+1}{2}} - (0.565)^{A+1} \right] dt$$

When integrating over equatorial pitch-angles $\alpha_0$, Liouville's theorem and the conservation of
the first adiabatic invariant ($\mu$) are taken into account: $\sin^2\alpha_{01} = B_0(L)/B(L,\lambda_0)$ and $\sin^2\alpha_{02} = $
$B_0(L)/B(L,\lambda)$, where $B(L,0) = B_0(L)$.
With an increase $\lambda$ from 0 to $\lambda_0 = 30^o$, the value of the function $\sqrt{4 - 3t^2}$ increases from 1 to
1.32, i.e. deviates from the average value (1.16) by only 16%. Most part of the ERB protons are



concentrated at these latitudes. Therefore, when calculating the last integral, we will assume that
$\sqrt{4-3t^2} \approx 1.16$.
Then you can get the following expression:
$$\Delta N(L, f_d) = k \, \frac{J(L, E(L, f_d))}{\sqrt{E(L, f_d)}} \, F(A) \, L^2 \, dL \, dE \ ,$$
where

$$F(A) = \frac{1}{A+1}\left[\frac{(1.16)^{-(A+1)/2}}{3A+11}\left(1-0.21\cdot0.65^A\right) - 0.085\,(0.565)^{A+1}\right]$$
and
$$k = (4\pi)^2 \, R_E^3 \, \sqrt{m/2} \ = 2.945\cdot10^{19} \ \text{cm}^2 \text{ s sr MeV}^{1/2}.$$
When calculating the values of $\Delta N$, we will take that $dL/L = dE/E = 0.1$. Finally, for the
indicated ERB region near the equatorial plane, we obtain:
$$\Delta N(L, f_d) = 2.945\cdot10^{17} J\left(L, E(L, f_d)\right)\sqrt{E(L, f_d)} \ F(A) \, L^3 \ , \tag{1}$$
where $J$, the differential fluxes of protons with equatorially pitch-angle $\alpha_0 \approx 90^\circ$, is given in units
of $(\text{cm}^2 \text{ s sr MeV})^{-1}$, and the energy of protons $E$ is given in MeV. The dependence $F(A)$ is shown
in Fig. 3.

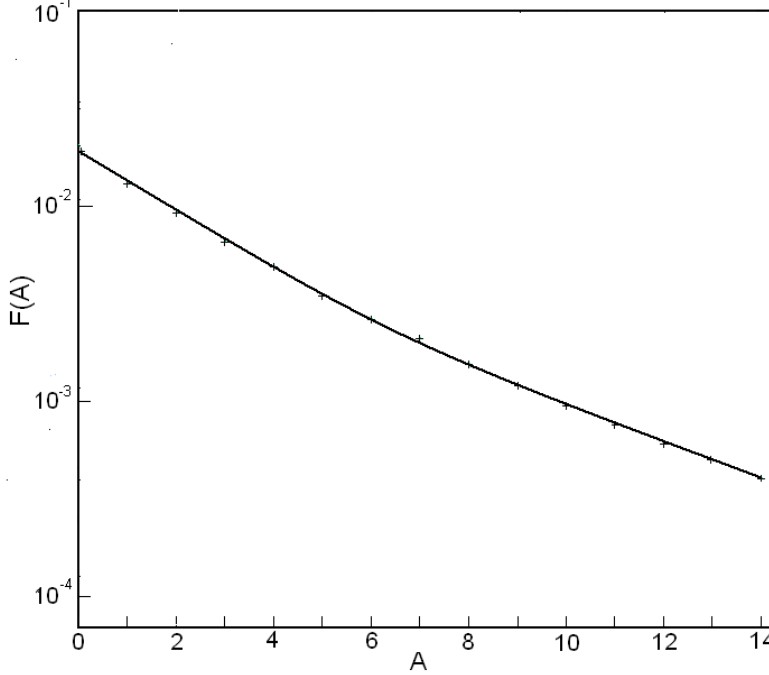

**Figure 3.** Dependence of the factor $F(A)$ in formula (1) on the anisotropy index $A$ of the proton fluxes.
For protons of the ERB, the radial profiles $\Delta N(L, f_d)$ for $f_d$ = 0.2, 0.3, 0.5, 1, 2, 3, 5. 10, 20, and
30 mHz, calculated by the formula (1) with using Figs. 1–3 are shown in Fig. 4, and the frequency
spectra $\Delta N(f_d, L)$ at $L$ = 2, 2.5, 3, 4, 5, and 6 are shown in Fig. 5. Near each curve in Fig. 4, the





corresponding value of $f_d$(mHz) is indicated, and each spectrum in Fig. 5 have the corresponding $L$
value (these values are highlighted in red). For clarity, in Figs. 4 and 5, thin curves alternate with
thick curves and in Fig. 5 spectra at $L = 2$ and 2.5 are highlighted in red.

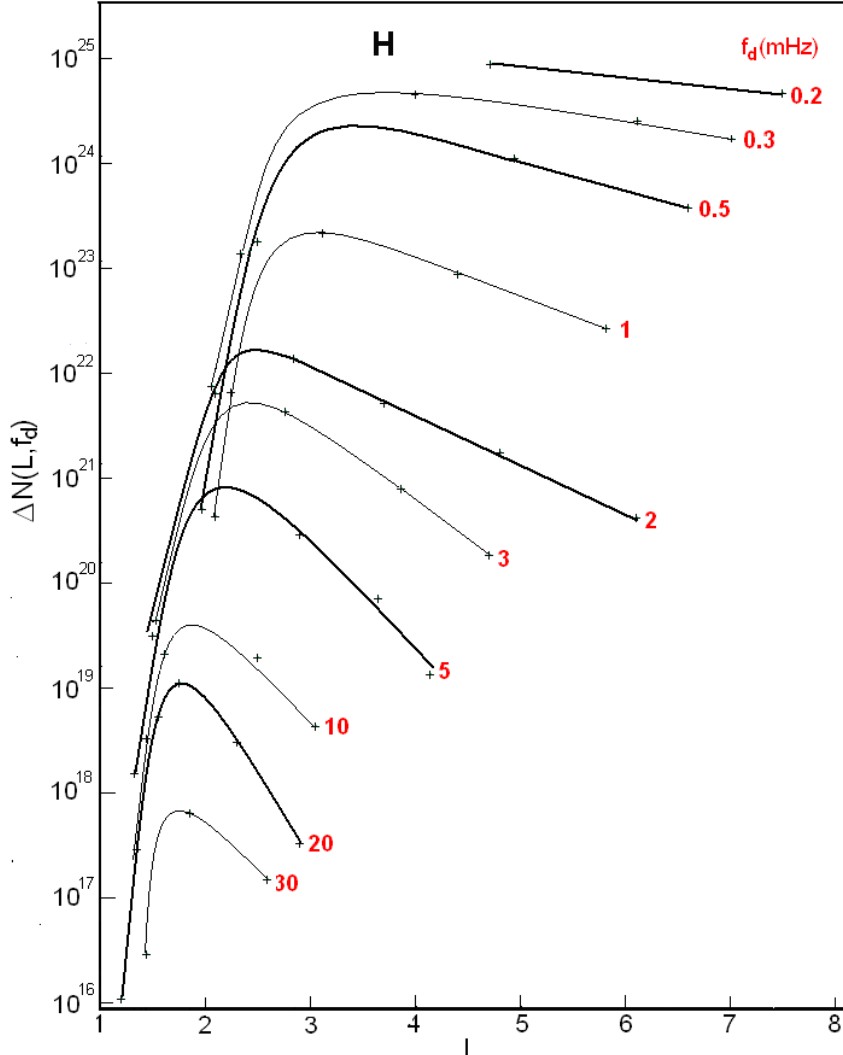


**Figure 4.** Radial profiles $\Delta N(L, f_d)$ for protons of the ERB with drift frequencies $f_d$ = 0.2, 0.3, 0.5, 1, 2, 3, 5. 10, 20 and

30 mHz, plotted for periods of solar activity maxima. The $f_d$ values corresponding to each curve are highlighted in red.
For clarity, thin curves are interspersed with thick curves.



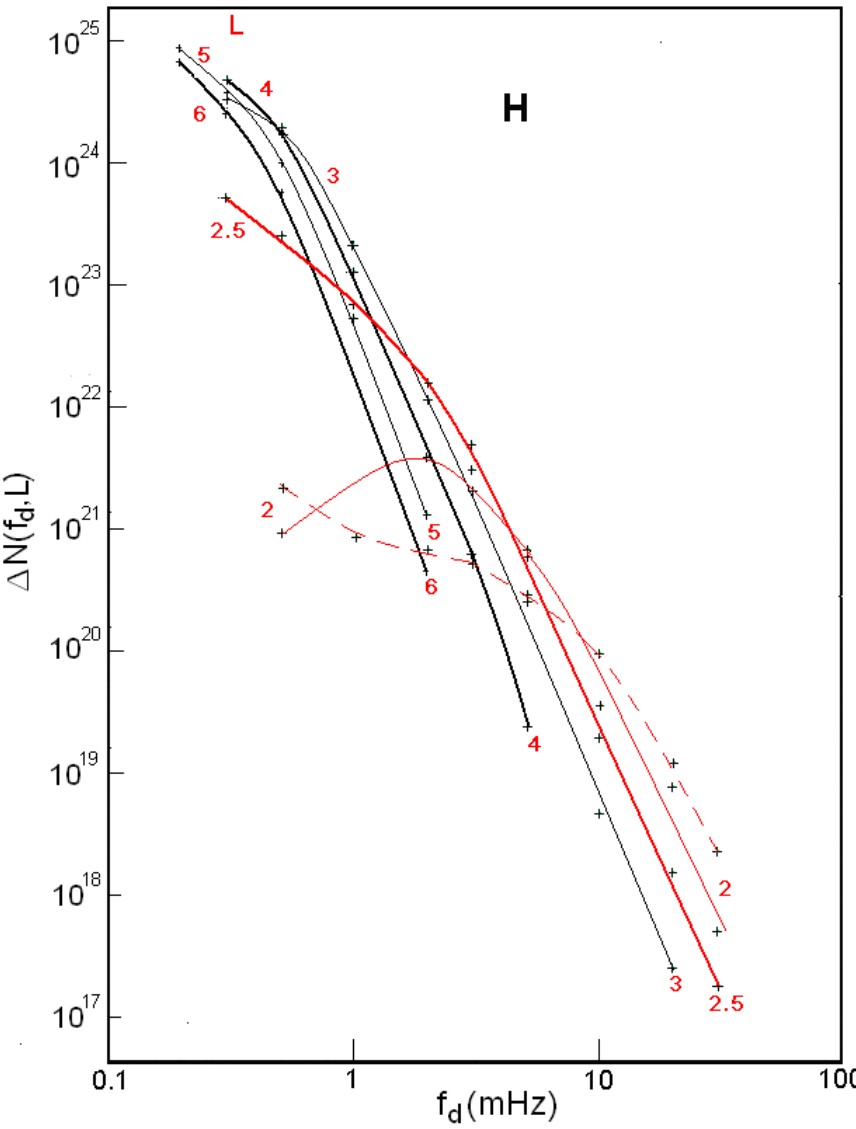

**Figure 5.** Frequency spectra $\Delta N(f_d, L)$ for protons of the ERB at $L = 2, 2.5, 3, 4, 5$ and 6, plotted for periods of solar activity maxima. The values $L$ corresponding to each spectrum and spectra at $L = 2$ and 2.5 are highlighted in red. The red dotted line shows the spectrum $\Delta N(f_d, L)$ of the ERB protons at $L = 2$, constructed from data for periods of solar activity minima (see Kovtyukh, 2020). For clarity, thin curves are interspersed with thick curves.

The errors of these calculations consist mainly of the errors of the averaged experimental data shown in Figs. 1 and 2 (these errors are most significant at $L < 2$), and because of the deviations of the geomagnetic field from the dipole model at $L > 5$.

As $\lambda_0$ decreases, the errors in our calculations will decrease. These errors can be reduced also by using numerical computer calculations. However, it should be taken into account that the fluxes of the ERB protons, as well as the energy spectra and pitch-angle distributions of these fluxes, may experience changes that exceed the errors of our calculations even in very quiet periods of observations.

**3 Discussion**





In agreement with the results of experimental and theoretical studies, at $L > 2.5$, the main
mechanism for the formation of the ERB for protons is the radial diffusion of particles from the
outer boundary of the geomagnetic trap to the Earth under conservation the adiabatic invariants $\mu$
and $K$ (see, e.g., Lejosne and Kollmann, 2020; Kovtyukh, 2016b, 2018).
Figures 1 and 2 presented here make it possible to determine in which regions of the space
$\{E,L\}$ near the equatorial plane the ionization losses of ions during their radial diffusion can be
neglected and where this cannot.
Iso-lines of the proton fluxes in Fig. 1 at sufficiently large $E$ and $L$ go up with decreasing $L$, in
the direction of increasing energy, in strict agreement with the adiabatic laws of radial transport of
particles. At lower $L$ these iso-lines reject to the low energies, under the influence of ionization
losses, which increase rapidly with decreasing $L$ (see in Kovtyukh, 2020 for details).
At sufficiently large values of $E$ and $L$, iso-lines of the anisotropy index in Fig. 2 pass
practically parallel to each other and parallel to the energy axis, in agreement with the laws of
adiabatic transport of particles with power-law energy spectra (see Kovtyukh, 1993). At lower $E$
and $L$, a more complex picture is formed under the influence of ionization losses (for more details
see in Kovtyukh, 2001, 2018).
With decreasing $L$, the radial diffusion are decreased very rapidly, and the belt of protons with $E$
$> 10–20$ MeV on $L < 2$ is generated mainly as result of decay a neutrons of albedo which are
knocked from the atmospheric atoms nuclei by the Galactic Cosmic Rays (GCR) protons. This
mechanism (CRAND) is simulated in many contemporary studies based on the experimental data
(see, e. g., Selesnick et al., 2007, 2013, 2014, 2018).
The mechanisms of formation of the ERB under the action of radial diffusion and CRAND are
manifested and clearly differ both in the radial profiles $\Delta N(L, f_d)$ and in the frequency spectra
$\Delta N(f_d, L)$ of protons.
Let us consider the manifestations of these mechanisms in Fig. 4 and 5 and related effects.
In contrast to the radial profiles of fluxes $J(L, E)$, the radial profiles $\Delta N(L, f_d)$ for protons with $f_d$
$> 10$ mHz (see Fig. 4) have much less steep of the outer edges and their steepness decreases with
decreasing frequency $f_d$. This effect is connected mainly with an increase in the volume of
magnetic tubes (factor $L^3$ in formula (1) from Section 2.3) and with a decrease in the anisotropy
index of proton fluxes with increasing $L$.
At the same time, in comparison with the radial profiles $J(L, E)$, the radial profiles $\Delta N(L, f_d)$
have more steeper inner edges. This effect is connect mainly with the large anisotropy of proton
fluxes in the corresponding region of space $\{E, L\}$ and with the rapid growth of the anisotropy
index with decreasing $L$ in this region. It is especially expressed in the radial profiles $\Delta N(L, f_d)$ at $f_d$
$\sim 0.3–1$ mHz (see Fig. 4); this is due to the fact that in the corresponding region of space $\{E, L\}$ the
anisotropy index of proton fluxes strongly depends on $E$ and $L$ (see Fig. 2).
Radial profiles $\Delta N(L, f_d)$ at $f_d > 10$ mHz are formed by the mechanism CRAND. They have a
maximum at $L \sim 1.5–2.0$, and the steepness of their inner and outer edges does not differ as much
as for lower frequencies $f_d$ (see Fig. 4). When constructing these profiles, it was taken into account
that at $E = 5–50$ MeV an anisotropy index $A$ of proton fluxes do not depend on $L$ at $L = 1.25–1.40$:
$A = 12\pm2$ (Fischer et al., 1977; Leonov et al., 2005).
The shape of the spectra $\Delta N(f_d, L)$ at $L > 3$ is determined, first of all, by the shape of the energy
spectra of proton fluxes $J(E, L)$ at the outer boundary of the geomagnetic trap. Gradually, as the
particles diffuse to the Earth, their energy spectra are transformed under the action of betatron
acceleration and ionization losses of particles.
In contrast to the energy spectra of proton fluxes $J(E, L)$, distributions $\Delta N(f_d, L)$ of the ERB
protons over their drift frequency $f_d$ (Fig. 5) differ much less from each other at $L > 3$. Such
convergence of the spectra $\Delta N(f_d, L)$ driven by increase in the volume of magnetic tubes and a



decrease in the anisotropy index of the ERB proton fluxes with increasing $L$. Figure 5 testify for
close to adiabatic transformations of the spectra $\Delta N(f_d, L)$ when $L$ changes at $L > 3$.
The energy spectra of near-equatorial proton fluxes $J(E, L)$ with $E > 10 \cdot L^{-3}$ MeV at $L > 3$ in
quiet periods have a local maximum at $E = (17\pm3) \cdot L^{-3}$ MeV and a power-law tail ($J \propto E^{-\gamma}$, where $\gamma$
$= 4.25\pm0.75$) at $E > (36\pm11) \cdot L^{-3}$ (Kovtyukh, 2001, 2018, 2020).
The frequency spectra of the ERB protons at $L > 3$ weakly depend on $L$ and over the considered
range $\Delta f_d$ have a close to power-law shape with an exponent $\gamma = 4.71\pm0.43$ (at $f_d > f_d^*$, where $f_d^*$
$\sim 0.5$ mHz at $L \sim 3$–6, $\sim 2$ mHz at $L = 2.5$ and $\sim 5$ mHz at $L = 2$). Note that the spread of the
parameter $\gamma$ for the frequency spectra of protons is almost 2 times less than for their energy spectra.
These spectra become more rigid (flattened) at $f_d < f_d^*$.
Thus, the average exponents of the power-law tail of the energy and frequency spectra of
protons differ by $\Delta\gamma = 0.46$, and there is no local maximum in the frequency spectra at $f_d > 2$ mHz
at $L > 2.5$. The main role in such differences in the shape of the energy and frequency spectra of
protons was played by the factor $F(A)$ in formula (1), in which the anisotropy index $A$ is a function
of $E$ and $L$ (see Figs. 2 and 3). Note that in the region $\{E > 0.5$ MeV, $L > 3\}$ the anisotropy index
$A$, as well as the protons energy, is transformed according to adiabatic laws when $L$ changes (see
Fig. 2 and comments to it).
These results confirm our hypothesis about the ordering of the distributions of protons over
their drift frequency $f_d$ in the outer regions of the ERB, at $L > 3$, where most of the ERB protons
are located and where the radial diffusion of protons overpowers their ionization losses.
At all $L$, the frequency spectra $\Delta N(f_d, L)$ become more flat at small $f_d$ and $E$ under influence
ionization losses. However, in the range of high $f_d$ (from 3–5 mHz to 30 mHz), for protons with
high energies and low ionization losses, the protons frequency spectra save a power-law tail even
at $L = 2$ (see Fig. 5).
For protons with $f_d < 0.5$ mHz, which correspond to the ERB protons of the lowest energies,
ionization losses lead to the same consequences at higher $L$-shells: the radial profiles $\Delta N(L, f_d)$
approach each other, and the spectra $\Delta N(f_d, L)$ flatten out (see Figs. 4 and 5).
In the region of the steep inner edge of the radial distributions $\Delta N(L, f_d)$, spectra $\Delta N(f_d, L)$ of the
ERB protons gradually, with decreasing $L$, become increasingly rigid and rapidly diverge from
each other (see Fig. 4 and 5). In the range of small $f_d$ at $L < 2.5$, the connection between these
distributions and the shape of the boundary energy spectra of protons is gradually lost.
These results indicate a violation of the order in the distributions of protons under the influence
of ionization losses.
In Fig. 5, the dotted line also shows the spectrum $\Delta N(f_d, L)$ of the ERB protons at $L = 2$,
constructed from experimental data for periods of low solar activity (see Fig. 1 in Kovtyukh,
2020). Figure 5 show that at $L = 2$ for $f_d > 10$ mHz there were more protons at the minimum of
solar activity, and for $f_d \sim 1$–10 mHz there were more protons at the maximum of solar activity.
The effect of a decrease in the $\Delta N(f_d, L)$ values for protons with $f_d > 10$ mHz at $L < 2$ with an
increase in solar activity is mainly connected with a decrease in the fluxes of protons with $E > 10$–
20 MeV here. This effect is well known. It is described by the CRAND mechanism (see, e.g.,
Selesnick et al., 2007) and was considered in detail in (Kovtyukh, 2020). With an increase in solar
activity, the densities of atmospheric atoms and ionospheric plasma on small $L$-shells significantly
increase, which leads to an increase in ionization losses of the ERB protons, but the power of their
main source (CRAND) practically does not change. As a result, the equilibrium fluxes and $\Delta N(f_d,$
$L)$ for protons with $f_d > 10$ mHz are establish at lower levels.



However, the effect of an increase in $\Delta N(f_d, L)$ for $f_d \sim 1$–10 mHz at low $L$ with increasing solar
activity, corresponding to the protons of lower energies, was discovered here for the first time.
With decreasing in $E$ (and $f_d$) of protons their ionization losses increase, and if the fluxes of
low-energy protons in the inner belt were also formed by the CRAND mechanism, one would have
observe even stronger increase of their fluxes with solar activity decreasing, than for protons with
$E > 10$–20 MeV ($f_d > 10$ mHz). But for protons with $f_d \sim 1$–10 mHz, we see in fig. 5 reverse effect
in the spectra $\Delta N(f_d, L)$ at $L = 2$, which is not described by the CRAND mechanism.
On the other hand, it was proved that stationary fluxes of protons with $E < 15$ MeV at $L \sim 2$ are
formed mainly by the mechanism of protons radial diffusion from the external region of the ERB
(Selesnick et al., 2007, 2013, 2014, 2018). These fluxes and $\Delta N(f_d, L)$ values for $f_d \sim 1$–10 mHz at
$L = 2$ are formed as a result of a balance of competing processes radial diffusion of protons and
their ionization losses.
The rates of transport of the ERB protons to the Earth (radial diffusion) rapidly increase with
decreasing particles energy (see Kovtyukh, 1916b). In addition, with an increase in solar activity,
the average level of geomagnetic fluctuations in the ERB increases. Under influence of these
factors, one can expect a significant increase in the intensity of radial diffusion of the low-energy
protons at low $L$ with an increase in solar activity. As a result, the effect of increasing in the
density of a dissipative medium with an increase in solar activity overpowered by a more
significant effect of increasing in the rates of radial diffusion of protons.
According to a numerous experimental data, during magnetic storms complex and varied
spectra of powerful pulsations of magnetic and electric fields in the frequency range considered
here (ULF) can be generate in the geomagnetic trap, which are non-regular distributed over $L$;
these pulsations can lead to local acceleration and losses of the ERB particles (see, e.g., Sauvaud et
al., 2013). Such effects will violate the regular character of the protons distributions shown in Fig.
4 and 5. However, in quiet periods, the amplitudes of such pulsations are small and they lead only
to radial diffusion of particles.

## 340    4 Conclusions

On the basis of generalized data on the fluxes of near-equatorial protons of the ERB with energy
from $E \sim 0.2$ MeV to 100 MeV at drift shells $L$ from $\sim 1$ to 8, the stationary distributions of the
ERB protons over the drift frequency of particles around the Earth ($f_d$) were constructed. The
results of calculations of the number of protons $\Delta N$ of the ERB within $30^o$ in geomagnetic latitude
at different $L$ and $f_d$ for periods of solar activity maximum are presented. They differ from the
corresponding distributions of the ERB protons for periods of low solar activity only at $L < 2.5$ (for
comparison, the spectra of these distributions are given at $L = 2$).
The radial profiles of these distributions $\Delta N(L, f_d)$ have one maximum that shift toward the
Earth with increasing $f_d$. In compare to the proton fluxes profiles $J(L, E)$, the radial profiles $\Delta N(L,$
$f_d)$ at $f_d < 5$ mHz have steeper inner edges and flatter outer edges. However, the radial profiles
$\Delta N(L, f_d)$ at $f_d > 10$ mHz, which are formed by the CRAND mechanism, have inner and outer edges
with only slightly difference from each other in the steepness.
In contrast to the energy spectra of proton fluxes $J(E, L)$, the frequency spectra $\Delta N(f_d, L)$ of the
ERB protons at $L > 3$ are weakly depend on $L$ and, for sufficiently large $f_d$ they have a nearly
power-law form with an exponent $\gamma = 4.71 \pm 0.43$. There is no local maximum in these spectra in
the region $\{f_d > 2$ mHz, $L > 2.5\}$, as in the corresponding $J(E, L)$ spectra.
Distributions $\Delta N(L, f_d)$ and $\Delta N(f_d, L)$ of the ERB protons in the region $\{f_d > 0.5$ mHz, $L > 3\}$
have a more orderly form than in the corresponding region of the space $\{E, L\}$, and the main
physical processes in the ERB manifested more clearly in these distributions. In these region most





of the ERB protons are located and the radial diffusion of protons overpowers their ionization
losses during the transport of particles to the Earth.
In the region of the steep inner edges of the radial distributions $\Delta N(L, f_d)$, the spectra $\Delta N(f_d, L)$
of protons rapidly diverge from each other with decreasing $L$, and at low frequencies these spectra
become flatten. These results indicate a violation of the order in these distributions of protons
under the influence of ionization losses.
With increasing in solar activity, the number of protons $\Delta N(f_d, L)$ at $L \sim 2$ decreases for $f_d > 10$
mHz and increases for $f_d \sim 1$–10 mHz. The effect at high $f_d$, corresponding to protons with $E > 15$
MeV, is well known and is described in the framework of the CRAND mechanism.
However, the opposite effect, at low $f_d$ corresponding to the lower-energy protons, is discovered
here for the first time. This effect can be associated with the fact that the low-frequency part of the
spectrum $\Delta N(f_d, L)$ of protons, even at $L \sim 2$, is formed mainly by the mechanism of protons
transport from the outer regions of the ERB. This effect may indicate that with increasing of the
solar activity, the average rates of radial diffusion of protons increase also. For low-energy protons
at $L \sim 2$, the effect of increasing density of a dissipative medium with increasing solar activity
overpowered by increasing the rates of radial diffusion of particles.

*Data availability*. All data from this investigation are presented in Figs. 1–5.
*Competing interests*. The author declares that there is no conflict of interest.
*Acknowledgements*. The author would like to thank the reviewers.






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
