# Peer review of "Distribution of Earth's radiation belts protons over the drift frequency of particles"

_Annales Geophysicae, 2020_

## Referee Comment (RC1) · Anonymous Referee #1 · 10 Nov 2020

General Comments

Dear author, I think that overall the paper itself is rather good. The topic introduced in this work is interesting and there is some novelty. The work has been conducted with care and all the calculations are explained in a clear way. Figures are well described in their captions and each one is recalled in the text. The description of the main body of the paper is a bit difficult in some points and they may not be easily followed by the reader. I propose a series of technical corrections that may help smooth the text a bit and make it a little easier to read. On the other hand, the conclusions are clearly explained and the discussion is rich. The references in the bibliography are targeted and complete.

Specific Comments

[Figure]

No specific comments

Technical Corrections

Line 1 Distribution of Earth's radiation belts protons . . .

Line 6-10 Thanks to the data on the proton fluxes of the Earth's radiation belts (ERB), with energy ranging from 0.2 to 100 MeV and drift L shell ranging from 1 to 8), their stationary distributions over the drift . . . are constructed. For this purpose, direct measurements of proton fluxes of the ERB in the period 1961–2017 near the geomagnetic equator were employed.

Line 12 . . . and their distributions in the space . . . have a more regular shape than . . .

Line 16 . . . is disrupted in advantage of transport . . .

Line 17 . . . with increasing solar activity, overpowers

Line 33 For the near-equatorial ERB protons, we have:

Line 36-37 with increasing amplitude of particles oscillation

Line 38-55 The frequency fc is different for different L shells (near the equatorial plane) and, as L increases (higher geomagnetic latitudes) the number of particles become less and less significant. For each given value of the frequency fb if L increases, then particles become more and more energetic (. . .) and their number becomes smaller . Compared to the frequencies fc and fb, the drift frequency fd for one particle species has a narrower range of values; it does not depend on the mass of the particles and it very weakly depends on the amplitude of their oscillations (. . .); in this case, on each L shell there are a significant number of particles corresponding to a certain value of fd . Therefore, it can be expected that the distributions of the ERB particles in the space {fd, L} will have a more regular shape than in the space {E, L}, and the main physical processes in these belts will manifest themselves more clearly in these distributions. Furthermore, it can be expected that on these more ordered background more fine

features can be revealed that would not appear in the space {E, L}. Despite the importance of the drift frequency fd for the mechanisms of the ERB formation, reliable and sufficiently complete distributions of particles in the ERBs (over the frequency fd ) have not been presented nor analyzed; indeed, this is the first time.

Line 56-61 The analysis presented in this paper is limited to the protons of the ERB during magnetically quiet periods of observations, when the proton fluxes and their spatial energy distributions were stationary. In the following sections, the distributions of the ERB protons over their drift frequency fd are constructed from experimental data (Sect. 2) and analyzed (Sect. 3). Finally, the main conclusions of this work are given in Sect. 4.

Line 67 In my opinion the term generalized is out of context here (and in similar statements)

Line 67-75 From the data of averaged satellite measurements of the differential fluxes of protons with an equatorial pitch-angle . . .., the aforementioned distributions are constructed in (Kovtyukh, 2020) during quiet periods. Such distributions, separated between periods near minima and maxima of the 11-year solar activity cycle, are constructed from satellite data also for other ionic components of the ERB (near the equatorial plane), but the most reliable and detailed picture was obtained in for protons (see Kovtyukh, 2020). In Fig. 1 one of these distributions is reproduced for periods near solar maxima (from 1968 to 2017); here, data of different satellites are associated with different symbols.

Line 77 correspond to the . . .

Line 83-84 The red lines correspond to the drift . . .

Line 85-97 Only protons with energies less than some maximum values, determined by the Alfvén's criterion: . . .. plane) can be trapped on the drift shells .

Line 91 The distribution of the ERB proton fluxes shown in Fig. 1, refers to ...

Interactive
comment

Line 109 . . . of these fluxes . . .

Line 111 . . . as red numbers.

Line 119 Figure 2 was written as Fig. 2 before.

Line 122 . . . energy-independent . . .

Line 123 in Fig. 2 are due to the fact . . .

Line 129 each other and to the energy axis . . .

Line 129 refers to protons . . .

Line 138 motions (these issues were most fully . . .

Line 139-140 Both the local maximum at . . . and the region of low anisotropy at . . . in Fig. 2, are related to the ionization losses of protons.

Line 145 which were obtained at . . .

Line 157 I believe that the unit of measurements for the B field is Gauss, G not Gs

Line 161 . . . , it increases by only . . .

Line 198 calculated using the formula (1) together with Figs . . .

Line 205 of maximum solar activity . . .

Line 208 see Line 205

Line 211 during minimum periods of solar activity

Line 222 of the ERB protons is the radial . . .

Line 225 Figs 1 and 2 . . .

Line 228 The iso-lines of proton fluxes in Fig. 1 at sufficiently large E . . .

Line 230 The use of the verb "to reject" is extremely unclear here, please clarify

Line 237 the radial diffusion is decreased very rapidly . . .

Line 247 . . . have much less steeper outer edges and . . .

Line 252 This effect is mainly connected to the large . . .

Line 268 . . . is driven by increase in the . . .

Line 270 This sentence here is rather unclear. Maybe it would be better to put it like: "Fig. 5 demonstrates the closeness to the adiabatic transformations of the spectra . . ."

Line 291 have a power-law tail . . .

Line 297 . . . become gradually increasingly rigid with decreasing L, and . . .

Line 304 Fig. 5 show that at . . .

Line 311 The word "but" here seems to be written with a smaller font with respect to "protons" and "the power"

Line 313 are established at lower . . .

Line 316 With decreasing E (and . . .

Line 318 observed

Line 318 with decreasing solar activity . . .

Line 319 . . . we see in Fig. 5 the opposite effect

Line 318 Under the influence of . . .

Line 331 . . . activity is overpowered by a more . . .

Line 333-339 According to numerous experimental data, during magnetic storms, a wide variety of complex spectra of powerful pulsations of magnetic and electric fields in the considered frequency range (ULF) can be generate in the geomagnetic trap, which are non-regularly distributed over L; these pulsations can lead to local acceleration and

losses of the ERB particles (...). Such effects will violate the regular characteristics of the protons distributions shown in Fig. 4 and 5. However, during quiet periods, the amplitudes of such pulsations are small and they lead only to radial diffusion of particles.

Line 341-343 Starting from the data on near-equatorial ERB proton fluxes (with energy ranging from 0.2 to 100 MeV and drift L shell ranging from 1 to 8), their stationary distributions . . . were constructed.

Line 344 . . . of the ERB protons within . . .

Line 345 . . . for periods of maximum solar activity . . .

Line 348-352 . . . have only one maximum that shifts toward . . . In comparison to the proton fluxes ... have steeper inner edges and flatter outer edges. However ... have inner and outer edges with only slightly difference from each other for what concerns the steepness of their profiles.

Line 354 . . . are weakly dependent on . . .

Line 355 power-law shape . . .

Line 358 have a more regular shape than . . .

Line 359 In these regions, there is the majority of the ERB protons, and their radial diffusion overpowers . . .

Line 366 With increasing solar activity, the number of protons . . .

Line 371 . . . is mainly formed by the mechanism . . .

Line 372-373 . . . that with increasing solar activity, the average rates of radial diffusion of protons increase as well.

Line 374-375 . . . with increasing solar activity is overpowered by the increase of the rates of radial . . .

---

## Referee Comment (RC2) · Anonymous Referee #2 · 30 Nov 2020

Review of "Distribution of the Earth's radiation belts protons over the drift frequency of particles" by Alexander S. Kovtyukh.

It is an interesting paper. It shows that the protons are bettered ordered by (Fd,L) then by (E,L) using data over a long period of time. The ERB protons try to conserve the flux invariant Ø in their drift orbit around the Earth. In a dipole field the drift frequency Td is proportional to E*L or E/Ø. That is that for a fixed energy E the drift frequency is proportional to the inverse of the flux invariant. The drift frequency is proportional to to the energy E. I recommend the paper to be published in Annales Geophyiiae taken into account the comments below.

Some comments:

[Figure]

1 Introduction The English in the paper should be improved line 85(as an example) On the drift shells only protons . . . . . . . . . . . . . .(near the equatorial plane) can be trapped.

2.2 Spatial-energy distribution . . . . . . . . . . . . . . OK

2.3 Drift . . . . . . . . . . . . . . It is more common to use j for differential flux than J , but OK OK

3 Discussion The discussion part of the paper is somewhat long and should be shorten. There is a lack of references to older work.

4 Conclusions Good

On the figures. Figure 1. Good figure. It exhibits how the protons are ordered in (E,L) space. Data from some of the satellites measure particles in and near the loss cone. How are these measurements transformed to particles mirroring at the equator should be explained. Figure 2. The anisotropy factor A should be defined. Figure 3 OK Figure 4 OK Figure 5 OK

―――――――――――――――――――――

---

## Author Comment (AC5) · 2 Dec 2020

Reply to Interactive comment by Anonymous Referee #2 from 30 November 2020 on the manuscript "Distribution of the Earth's radiation belts protons over the drift frequency of particles" by Alexander S. Kovtyukh

Deeply respected Referee #2,

I am very grateful to you for this review. All your comments are very helpful for me and it is taken into account in the manuscript. In the last version of the manuscript, the text

has been carefully checked and aligned with the rules of English grammar. Corrections made by Referee # 1 highlighted in blue. This revised version of the article contains several clarifying notes (Lines 137-143, 362-363, 383-386), which are highlighted in green.

With grand regard, Alexander S. Kovtyukh

Some comments:

1 Introduction The English in the paper should be improved line 85(as an example) On the drift shells only protons . . .. . .. . .. . ...(near the equatorial plane) can be trapped. AC: I agree. Text corrected. 2.2 Spatial-energy distribution . . .. . .. . .. . .. . .. OK 2.3 Drift . . .. . .. . .. . .. . ... It is more common to use j for differential flux than J , but OK AC: I agree. I also use the letter j to denote differential particle fluxes, but in this, as in the previous article, I use the letter J. Please save it here. 3 Discussion The discussion part of the paper is somewhat long and should be shorten. There is a lack of references to older work. AC: I tried to write this part of the article as briefly as possible, excluding everything secondary. The first four paragraphs of Sect. 3 could be moved to the end of Sect. 2.2, but I think they are more appropriate here. A very detailed list of works on the ERB, starting from 1961, is presented in the review (Kovtyukh, 2018) in Space Sci. Rev., but here only the most important works are given (some of the most important old works added to Lines 110-115). 4 Conclusions Good On the figures. Figure 1. Good figure. It exhibits how the protons are ordered in (E,L) space. Data from some of the satellites measure particles in and near the loss cone. How are these measurements transformed to particles mirroring at the equator should be explained. AC: Some of the data shown in Fig. 1 were obtained on polar satellites (Injun-5, OV1-19 and Azur), but in Fig. 1 (Kovtyukh, 2020) we use only data of these satellites at L < 1.6 which were obtained near the equatorial plane. Figure 2. The anisotropy factor A should be defined. AC: Here, the generally accepted definition of the anisotropy index of the ERB particle fluxes is used (Lines 95-97). At small equatorial pitch angles, these distributions may have a more complex shape and cannot be described by one simple

parameter, but here we consider only particles with mirror points within 30 degrees in geomagnetic latitude.

---

## Author Comment (AC6) · 2 Dec 2020

The comment was uploaded in the form of a supplement:
https://angeo.copernicus.org/preprints/angeo-2020-67/angeo-2020-67-AC6-supplement.pdf
* * *

---

## Author Comment (AC7) · 2 Dec 2020

The comment was uploaded in the form of a supplement:
https://angeo.copernicus.org/preprints/angeo-2020-67/angeo-2020-67-AC7-supplement.pdf
* * *

---

## Author Comment (AC10) · 12 Dec 2020

Deeply respected Dr. Elias Roussos,

all comments of the reviewers have been taken into account, the text has been corrected (the final version of the manuscript is given in AC10). The complete response to Anonymous Referee #1 (RC1) is in AC1 (AC4), and the revised text is in AC2 (corrections) and AC3 (revised and complemented). The complete response to Anonymous Referee #2 (RC2) is in AC5 (AC9), and the revised text is in AC8. One extra article "the" has been removed from the title of the article.

With grand regard, Alexander S. Kovtyukh

I am very grateful to Referee #1 and Referee #2. I reproduce here my reply to these

reviews.

Deeply respected Referee #1,

I am very grateful to you for such an exclusively generous and thorough review. Thank you very much for these corrections! All these comments are very helpful for me and it is taken into account in the manuscript. I think that the manuscript now is much linear and easy to read. A paragraph has been added to Sect. 2, which explain as I estimate the anisotropy of proton fluxes at L > 6 (highlighted in green).

With grand regard, Alexander S. Kovtyukh

Deeply respected Referee #2,

I am very grateful to you for an exclusively generous and thorough review. All these comments are very helpful for me and it is taken into account in the manuscript. In the last version of the manuscript, the text has been carefully checked and aligned with the rules of English grammar. Corrections made by Referee #1 highlighted in blue. This revised version of the manuscript contains several clarifying notes (Lines 137-143, 362-363, 383-386), which are highlighted in green.

With grand regard, Alexander S. Kovtyukh

---

## Author Response (AR1)

**Topical Editor Decision: Publish subject to minor revisions (review by editor)**
26 Dec 2020) by Elias Roussos

**Comments to the Author**:

Dear Dr. Kovtyukh,
thank you for submitting your manuscript "Distribution of the Earth's radiation belts protons over the drift frequency of particles" to Ann. Geophys. As you may have realised from the two referee reports, available also in the interactive discussion, the two reviewers find your work of importance and appropriate for publication in our journal. They still, however, remark that there are numerous (minor) presentation issues that need to be resolved before we can proceed with the next stages of peer review. I appreciate your effort in providing already some insights on how these issues would be tackled (interactive discussion). From an editorial perspective, I would still need to see some more discussion on how data have been treated as well as more extensive references to earlier work, such that the originality of your methods and results can be highlighted. For example:

1) How did you perform the averaging of the datasets from different instruments, given that many have different detector characteristics and there may be cross-calibration issues? Did you use a different weight for different instruments that may obtain higher quality spectra?

AC: The problem of methodical differences in measurements of the fluxes of protons of the radiation belts on different satellites was one of the main ones in this work, as well as in the paper (Kovtyukh, 2020), of which it is partly a continuation. First, a correlation analysis and selection of published experimental data that are in good agreement with each other (for all satellites presented here) were carried out. In this case, we excluded from consideration all unreliable measurement results, which were pointed out by the authors of the cited works (the admixture of electrons and various ionic components of the ERB to the protons). Then, the reliable results of measurements (near the equatorial plane) of fluxes and anisotropy of proton fluxes obtained on these satellites were introduced into the space $\{E, L\}$, shown in Fig. 1 and 2. In such representation of experimental data, there is no need for interpolation and extrapolation of fluxes on the energy; in other representations, such necessity arises due to differences in channel widths and their positions on the energy scale for instruments installed on different satellites). In addition, for such representation, in one figure, the data of various experiments, it is possible to construct the isolines of fluxes (and anisotropy of fluxes); these isolines do not intersect with each other and, thus, allow to exclude results that sharply fall out of the general picture.
Text supplemented (lines 67-80).

Are all solar cycle phases covered with the same frequency, or do certain solar cycle phases dominate in the averaging?

AC: Figs. 1, 4 and 5 (except for the dotted spectrum for $L = 2$ in Fig. 5), are based on the measurements near solar activity maximum in 20th (1968–1971), 22th (1990–1991), 23th (2000), and 24th (2012–2017) solar cycles. The dotted line in Fig. 5 shows the frequency spectrum of the ERB protons at $L = 2$, which constructed from experimental data for periods near solar activity minima between the 19th/20th, 20th/21th, 21th/22th, and 22th/23th solar cycles.
Text supplemented (lines 88-89 and 343-344).

How do you define "quiet periods", is there a source/reference of your data (rather just saying "data from Van Allen Probes")? Some of this information is partly in your companion paper, but some more details would be useful here for the completeness of the work.

AC: In this work, only the experimental data obtained at Kp < 2 (under quiet conditions in the Earth's magnetosphere and in the heliosphere) were used. A complete list of works used in the construction of Figs. 1 and 5, contained in my previous article in Ann. Geophys. (Kovtyukh, 2020).
Text supplemented (lines 61, 88, 133-134, 380).

2) How representative is the averaging for high $L$-shells? Is there an impact (влияние) due to the fact that you use a dipole $L$ instead of an $L^*$?

AC: During quiet periods considered in this work, the geomagnetic field at $L < 5$-5.5 is close to the dipole configuration and $L \approx L^*$. At large $L$, the magnetic field differs from the dipole one even in quiet periods; it is leads to the flattening of the isolines of the proton fluxes at $L > 5$ in Fig. 1.
Text supplemented (lines 106-108).

3) Several papers by Selesnick et al. and others are referenced but barely discussed, e.g. aspects of solar energetic particle entry etc. Its difficult to see how the methodology introduced in this paper offers advantages in the interpretation of data compared to what has been done so far.

AC: Since I have considered here only ERB proton fluxes measured during quiet periods, the effects observed during strong magnetic storms ($Dst < –200$ nT), as well as the effects associated with solar flares, substorms, and other disturbances, are not considered here. It is only indicated that magnetic activity leads to a transformations of the distributions of protons (and other particles) of the ERB (lines 375-381).

I recommend that you prepare and formally submit your revised article along with a point-by-point answer to all the reviewer/editorial comments and a version of your revision with all changes marked-up. You may reuse the answers and material you already provided in the interactive discussion, but if you find that additional changes or clarifications are needed, feel free to include them. Your revision will likely be reviewed only by myself, unless, based on the quality of the revised article, I find it necessary to resend it for at least one more round of reviews.

AC: I accept all comments of the reviewers. It is take into account in the corrected and supplemented version of the manuscript. Thank you very much for your work.

Kind regards,
Alexander Kovtyukh

AC: Below are my responses to the reviews. These comments helped me very much to improve and clarify the manuscript.

*Reply to Interactive comment* **by Anonymous Referee #1 from 10 November 2020 on the manuscript "Distribution of the Earth's radiation belts protons over the drift frequency of particles"** *by* **Alexander S. Kovtyukh**

**General Comments**

Dear author, I think that overall the paper itself is rather good. The topic introduced in this work is interesting and there is some novelty. The work has been conducted with care and all the calculations are explained in a clear way. Figures are well described in their captions and each one is recalled in the text. The description of the main body of the paper is a bit difficult in some points and they may not be easily followed by the reader. I propose a series of technical corrections that may help smooth the text a bit and make it a little easier to read. On the other hand, the conclusions are clearly explained and the discussion is rich. The references in the bibliography are targeted and complete.

**Specific Comments**
No specific comments

**Technical Corrections**

Line 1 Distribution of Earth's radiation belts protons …
  AC: I agree. Text corrected.

Line 6-11 Thanks to the data on the proton fluxes of the Earth's radiation belts (ERB) with energy ranging from 0.2 to 100 MeV and drift $L$ shells ranging from 1 to 8, their stationary distributions over the drift frequency $f_d$ of protons around the Earth are constructed. For this purpose, direct measurements of proton fluxes of the ERB in the period 1961–2017 near the geomagnetic equator were employed.
  AC: I agree. Text corrected.

Line 12-13 … and their distributions in the space … have a more regular shape than …
  AC: I agree. Text corrected.

Line 17 … is disrupted in advantage of transport …
  AC: I agree. Text corrected.

Line 18 … with increasing solar activity, overpowers …

AC: I agree. Text corrected.

Line 33 For the near-equatorial ERB protons, we have:
AC: I agree. Text corrected.

Line 37 … with increasing amplitude of particles oscillation.
AC: I agree. Text corrected.

Line 38-59 The frequency $f_c$ is different for different $L$-shells (near the equatorial plane) and, as $L$ increases (higher geomagnetic latitudes) the number of particles become less and less insignificant. For each given value of the frequency $f_b$ if $L$ increases, then particles become more and more energetic (…) and their number becomes smaller.

Compared to the frequencies $f_c$ and $f_b$, the drift frequency $f_d$ for one particle species has a narrower range of values; it does not depend on the mass of the particles and it very weakly depends on the amplitude of their oscillations (…); in this case, on each $L$-shell there are a significant number of particles corresponding to a certain value of $f_d$.

Therefore, it can be expected that the distributions of the ERB particles in the space $\{f_d, L\}$ will have a more regular shape than in the space $\{E, L\}$, and the main physical processes in these belts will manifest themselves more clearly in these distributions. Furthermore, it can also be expected that on these more ordered background more fine features can be revealed that would not appear in the space $\{E, L\}$.

Despite the importance of the drift frequency $f_d$ for the mechanisms of the ERB formation, reliable and sufficiently complete distributions of particles in the ERBs (over the frequency $f_d$) have not been presented nor analyzed; indeed, this is the first time.
AC: I agree. Text corrected.

Line 60-65 The analysis presented in this paper is limited to the protons of the ERB during magnetically quiet periods of observations, when the proton fluxes and their spatial energy distributions were stationary. In the following sections, the distributions of the ERB protons over their drift frequency $f_d$ are constructed from experimental data (Sect. 2), and analyzed (Sect. 3). Finally, the main conclusions of this work are given in Sect. 4.
AC: I agree. Text corrected.

Line 85 In my opinion the term generalized is out of context here (and in similar statements).
AC: I agree. Text corrected.

Line 85-96 From the data of averaged satellite measurements of the differential fluxes of protons with an equatorial pitch-angle …, aforementioned distributions are constructed in (Kovtyukh, 2020) during quiet periods. Such distributions, separately between periods near minima and maxima of the 11-year solar activity cycle, are constructed from satellite data also for other ionic components of the ERB (near the equatorial plane), but the most reliable and detailed picture was obtained in for protons (see Kovtyukh, 2020). In Fig. 1 one of these distributions is reproduced for periods near solar maxima (from 1968 to 2017); here, data of different satellites are associated with different symbols.
AC: I agree. Text corrected.

Line 98 … correspond to the …

AC: I agree. Text corrected.

Line 98 The red lines correspond to the drift …
AC: I agree. Text corrected.

Line 109-112 Only protons with energies less than some maximum values, determined by the Alfvĕn's criterion: … plane) can be trapped on the drift shells.
AC: I agree. Text corrected.

Line 115 The distribution of the ERB proton fluxes shown in Fig. 1, refers to …
AC: I agree. Text corrected.

Line 125 … of these fluxes …
AC: I agree. Text corrected.

Line 131 … as red numbers.
AC: I agree. Text corrected.

Line 142 Figure 2 was written as Fig. 2 before.
AC: I agree. Text corrected.

Line 145 … energy-independent …
AC: I agree. Text corrected.

Line 146 … in Fig. 2 are due to the fact …
AC: I agree. Text corrected.

Line 152 … each other and to the energy axis …
AC: I agree. Text corrected.

Line 152 … refers to protons …
AC: I agree. Text corrected.

Line 166 … motions (these issues were most fully …
AC: I agree. Text corrected.

Line 167-168 Both the local maximum at … and the region of low anisotropy at … in Fig. 2, are related to the ionization losses of protons.
AC: I agree. Text corrected.

Line 180 … which were obtained at …
AC: I agree. Text corrected.

Line 192 I believe that the unit of measurements for the B field is Gauss, G not Gs.
AC: I agree. Text corrected.

Line 161 … , it increases by only …
AC: I agree. Text corrected.

Line 234 … calculated using the formula (1) together with Figs …
AC: I agree. Text corrected.

Line 241 … of maximum solar activity …
AC: I agree. Text corrected.

Line 245 see Line 205
AC: I agree. Text corrected.

Line 247 … during minimum periods of solar activity …
   AC: I agree. Text corrected.

Line 259 … of the ERB protons is the radial …
   AC: I agree. Text corrected.

Line 262 Figs. 1 and 2 …
   AC: I agree. Text corrected.

Line 265 The iso-lines of the proton fluxes in Fig. 1 at sufficiently large $E$ …
   AC: I agree. Text corrected.

Line 267-268 The use of the verb "to reject" is extremely unclear here, please clarify.
   AC: I agree. Text corrected.

Line 275 … the radial diffusion is decreased very rapidly …
   AC: I agree. Text corrected.

Line 285 … have much less steeper outer edges and …
   AC: I agree. Text corrected.

Line 290 This effect is mainly connected to the large …
   AC: I agree. Text corrected.

Line 306 … is driven by increase in the …
   AC: I agree. Text corrected.

Line 307-308 This sentence here is rather unclear. Maybe it would be better to put it like:
"Fig. 5 demonstrates the closeness to the adiabatic transformations of the spectra …
   AC: I agree. Text corrected.

Line 330 … have a power-law tail …
   AC: I agree. Text corrected.

Line 336 … become gradually increasingly rigid with decreasing $L$, and …
   AC: I agree. Text corrected.

Line 344 Fig. 5 show that at …
   AC: I agree. Text corrected.

Line 352 The word "but" here seems to be written with a smaller font with respect to
"protons" and "the power".
   AC: I agree. Text corrected.

Line 354 … are established at lower …
   AC: I agree. Text corrected.

Line 357 With decreasing $E$ (and …
   AC: I agree. Text corrected.

Line 359 … with decreasing solar activity …
   AC: I agree. Text corrected.

Line 361 … we see in Fig. 5 the opposite effect …
   AC: I agree. Text corrected.

Line 370 Under the influence of …

AC: I agree. Text corrected.

Line 373 … activity is overpowered by a more …
AC: I agree. Text corrected.

Line 375-381 According to numerous experimental data, during magnetic storms, a wide variety of complex spectra of powerful pulsations of magnetic and electric fields in the considered frequency range (ULF) can be generate in the geomagnetic trap, which are non-regularly distributed over $L$; these pulsations can lead to local acceleration and losses of the ERB particles (…). Such effects will violate the regular characteristics of the protons distributions shown in Fig. 4 and 5. However, during quiet periods, the amplitudes of such pulsations are small and they lead only to radial diffusion of particles.
AC: I agree. Text corrected.

Line 383-385 Starting from the data on near-equatorial ERB proton fluxes (with energy from 0.2 to 100 MeV and drift $L$ shells ranging from 1 to 8), their stationary distributions … were constructed.
AC: I agree. Text corrected.

Line 386-387 … of the ERB protons within …
AC: I agree. Text corrected.

Line 387-388 … for periods of maximum solar activity …
AC: I agree. Text corrected.

Line 391-396 … only one maximum that shifts toward… In comparison to the proton fluxes … have steeper inner edges and flatter outer edges. However … have inner and outer edges with only slightly difference from each other for what concerns the steepness of their profiles.
AC: I agree. Text corrected.

Line 398 … are weakly dependent on …
AC: I agree. Text corrected.

Line 399 … power-law shape …
AC: I agree. Text corrected.

Line 404 … have a more regular shape than …
AC: I agree. Text corrected.

Line 405-407 In these regions, there is the majority of the ERB protons, and their radial diffusion overpowers …
AC: I agree. Text corrected.

Line 412 With increasing solar activity, the number of protons …
AC: I agree. Text corrected.

Line 417 … is mainly formed by the mechanism …
AC: I agree. Text corrected.

Line 418-419 … that with increasing solar activity, the average rates of radial diffusion of protons increase as well.
AC: I agree. Text corrected.

Line 420-421 … with increasing solar activity is overpowered by the increase of the rates of radial …

AC: I agree. Text corrected.

Deeply respected Referee #1,
I am very grateful to you for such an exclusively generous and thorough review. Thank you very much for these corrections! All these comments are very helpful for me and it is taken into account in the manuscript. I think that the manuscript now is much linear and easy to read.
A paragraph has been added to Sect. 2, which explain as I estimate the anisotropy of proton fluxes at $L > 6$ (lines 169-175).
Kind regards,
Alexander Kovtyukh

**Reply to Interactive comment* by Anonymous Referee #2 from 30 November 2020 on the manuscript "Distribution of the Earth's radiation belts protons over the drift frequency of particles" *by* Alexander S. Kovtyukh**

It is an interesting paper. It shows that the protons are bettered ordered by $\{f_d, L\}$ then by $\{E, L\}$ using data over a long period of time. The ERB protons try to conserve the flux invariant $\Phi$ in their drift orbit around the Earth. In a dipole field the drift frequency $f_d$ is proportional to $EL$ or $E/\Phi$. That is that for a fixed energy $E$ the drift frequency is proportional to the inverse of the flux invariant. The drift frequency is proportional to the energy $E$. I recommend the paper to be published in Annales Geophysicae taken into account the comments below.

**Some comments:**

1 Introduction. The English in the paper should be improved, line 85 (as an example): On the drift shells only protons (near the equatorial plane) can be trapped.

AC: I agree. Text corrected.

2.2 Spatial-energy distribution. OK

2.3 It is more common to use $j$ for differential flux than $J$, but OK.

AC: I agree. I also use the letter $j$ to denote differential particle fluxes, but in this, as in the previous paper, I use the letter $J$. Please save it here.

3 Discussion The discussion part of the paper is somewhat long and should be shorten. There is a lack of references to older work.

AC: I tried to write this part of the article as briefly as possible, excluding everything secondary. The first four paragraphs of Sect. 3 could be moved to the end of Sect. 2.2, but I think they are more appropriate here. A very detailed list of works on the ERB, starting from 1961, is presented in the review (Kovtyukh, 2018) in Space Sci. Rev., but here only the most important works are given (some of the most important old works added to lines 141-146).

4 Conclusions Good

On the figures.

Figure 1. Good figure. It exhibits how the protons are ordered in ($E$, $L$) space. Data from some of the satellites measure particles in and near the loss cone. How are these measurements transformed to particles mirroring at the equator should be explained.

AC: Some of the data shown in Fig. 1 were obtained on polar satellites (Injun-5, OV1-19 and Azur), but in Fig. 1 (Kovtyukh, 2020) I use only data of these satellites at $L < 1.6$ which were obtained near the equatorial plane.

Figure 2. The anisotropy factor $A$ should be defined.

AC: Here, the generally accepted definition of the anisotropy index of the ERB particle fluxes is used (lines 122-126). At small equatorial pitch angles, these distributions may have a more complex shape and cannot be described by one simple parameter, but here we consider only particles with mirror points within 30 degrees in geomagnetic latitude.

Figure 3 OK

Figure 4 OK

Figure 5 OK

Deeply respected Referee #2,

I am very grateful to you for an exclusively generous and thorough review. All these comments are very helpful for me and it is taken into account in the manuscript. In the last version of the manuscript, the text has been carefully checked and aligned with the rules of English grammar. This revised version of the manuscript contains several clarifying notes (lines 67-80, 87-89, 106-108. 173-175, 344-345, 402-403, 423-426).

Kind regards,

Alexander Kovtyukh